# Impact of Co-Fermentation on the Soluble Pentosan, Total Phenol, Antioxidant Activity, and Flavor Properties of Wheat Bran

**DOI:** 10.3390/microorganisms13071546

**Published:** 2025-07-01

**Authors:** Yan Chen, Li Zhang, Yifan Chen, Hongling Wang, Wenpei Ge, Zhanying Xue, Xinran Cui, Xin Wang, Aimei Liao, Yuansen Hu, Na Liu

**Affiliations:** 1School of Biological Engineering, Henan University of Technology, 100 Lianhua Street, Zhengzhou 450001, China; 2024930700@stu.haut.edu.cn (Y.C.); zhanglibio@haut.edu.cn (L.Z.); yifanchen@stu.haut.edu.cn (Y.C.); 17864099663@163.com (H.W.); gwpp18790641140@163.com (W.G.); whitexue09@gmail.com (Z.X.); 17839922361@163.com (X.C.); wangxin@haut.edu.cn (X.W.); aimeiliao@haut.edu.cn (A.L.); hys308@126.com (Y.H.); 2Henan Provincial Key Laboratory of Grain Resources Conservation and Utilization, College of Biological Engineering, Henan University of Technology, Zhengzhou 450001, China; 3National Engineering Research Center of Wheat and Corn Further Processing, Henan University of Technology, Zhengzhou 450001, China

**Keywords:** wheat bran, co-fermentation, flavor properties, soluble pentosan, total phenol, antioxidant activity, *Lactiplantibacillus plantarum*, *Monascus purpureus*, GC-MS

## Abstract

Most wheat bran in China is used as animal feed due to its coarse taste. However, fermentation can degrade cellulose in wheat bran, improving its taste and flavor. The effects of co-fermentation with *Lactiplantibacillus plantarum* and *Monascus purpureus* on wheat bran remain unclear. In this study, we evaluated soluble pentosan, total phenol content, and antioxidant activity in co-fermented wheat bran. Following treatment, the soluble pentosan content was 6.03-fold higher than in raw bran. The total phenol content increased by 5.74-fold, and antioxidant activity was enhanced as well. The flavor profile of wheat bran co-fermented with *L. plantarum* and *M. purpureus* changed significantly, with alcohols and pyrazines increasing by 3- to 20-fold and aldehydes decreasing by 52.76%, resulting in a fruity, sweet, and nutty aroma. This study supports the comprehensive utilization of wheat bran and provides novel insights into improving its functionality and quality.

## 1. Introduction

China is the largest producer and consumer of wheat globally, generating at least 20 million tons of wheat bran (WB) annually [1]. As a major byproduct of wheat milling, WB is rich in protein, lipids, crude fiber, and starch and holds promise as a functional food ingredient. Despite its nutritional benefits and application potential, most WB is used as animal feed and remains underutilized. Therefore, it is essential to develop advanced processing technologies to enhance its added value. Fermentation offers an efficient means to convert low-value substrates into high-value-added products by generating various bioactive compounds. It enhances the nutritional profile of food and produces components that promote health while also improving flavor [2].

*M. purpureus* is a small filamentous saprophytic fungus that is typically inoculated into rice to produce fermented red yeast rice. Red yeast rice is a product commonly used in the production of wine and vinegar and is also a coloring and flavoring agent for fermented tofu [3]. However, fermentation with *Monascus purpureus* has been shown to increase phenolic acid content and antioxidant activity in rice bran [4]. Bedő et al. demonstrated the feasibility of producing arabinose and xylitol from WB through fermentation [5]. *Lactic acid bacteria* have also been reported to enhance WB bioactivity by increasing the total phenolic content, thereby improving antioxidant capacity and flavor [6]. *L. plantarum* is a safe and effective probiotic and is widely used in the medical, food, and feed industries [7]. In addition, due to the production of various bioactive components and functional factors during the fermentation process of *L. plantarum*, which have potential antioxidant and hypoglycemic functions, they have become increasingly popular in fermented food research in recent years [8]. The use of *yeast* strains and *lactic acid bacteria* (LAB) for the fermentation of mahali plum fruit is a sustainable method that has characteristics such as flavor improvement and texture enhancement [9]. Studies suggest that co-fermentation with *yeast* and *lactic acid bacteria* can substantially enhance the content and bioavailability of bioactive substances in wheat bran [10]. For instance, co-fermentation involving *M. purpureus, L. plantarum*, and Saccharomyces cerevisiae has been shown to affect free amino acid levels and flavor characteristics in Pyropia yezoensis [11]. However, limited research has investigated the combined effects of *Lactiplantibacillus* and *Monascus* fermentation on WB properties, and it remains unclear whether co-fermentation can enhance WB’s value. Notably, *Lactobacillus* fermentation contributes to the generation of complex flavor compounds, such as amino acids and organic acids, which improve the flavor and palatability of the fermentation broth [12]. Therefore, co-fermentation with *Lactiplantibacillus* and *Monascus* may be a promising strategy for improving WB’s flavor. Nonetheless, studies exploring the antioxidant potential and flavor profile of co-fermented WB are lacking. This study investigated changes in the soluble pentosan content, antioxidant activity, and volatile compounds in co-fermented WB, providing a foundation for its potential application in functional foods.

## 2. Materials and Methods

### 2.1. Materials, Reagents, and Strains

Wheat bran was provided by Henan Feitian Biotechnology Co., Ltd. (Hebi City, China). Chemicals and reagents were purchased from Shanghai McLean Biochemical Science and Technology Co., Ltd. (Shanghai, China). *Lactiplantibacillus plantarum* (HNCIMC: 24809) and *Monascus purpureus* (HNCIMC: 40944) were obtained from the Henan Provincial Engineering Laboratory of Preservation and Breeding of Industrial Microbial Strains, Henan University of Technology (Zhengzhou, China). *L. plantarum* was cultured in the Man Rogosa Sharpe (MRS) liquid medium, while *M. purpureus* was cultured in a potato dextrose agar (PDA) liquid medium.

### 2.2. Preparation of Microorganism Inoculum

The *L. plantarum* seed culture was prepared by incubating 50 mL of MRS medium in a 250 mL Erlenmeyer flask at 37 °C and 150 rpm for 28 h (logarithmic phase). The *M. purpureus* seed culture was prepared by punching agar blocks from PDA plates previously cultured at 28 °C for 4 days. The blocks were rinsed with sterile saline and adjusted to an OD_600_ of 0.6 for inoculum preparation.

### 2.3. Solid-State Fermentation of Wheat Bran

Extruded wheat bran was prepared as described by Roye [13] and adjusted to a moisture content of 60%. The substrate was sterilized by autoclaving at 115 °C for 20 min and cooled before inoculation. Sterilized wheat bran was inoculated with 10% (*v*/*w*) of the respective microbial inoculum and incubated at 30 °C for 0, 2, 4, 6, or 8 days in a constant-temperature shaker (150 rpm) (Shanghai Zhichu Instrument Co., Ltd., Shanghai, China). Control samples were treated identically but inoculated with 10 mL of sterile water per 100 g of wheat bran. All samples were dried at 50 °C and passed through a 40-mesh sieve prior to analysis.

Unfermented wheat bran was labeled as C-WB. Wheat bran fermented with *L. plantarum* alone was labeled as L-WB, wheat bran fermented with *M. purpureus* alone was labeled as M-WB, and wheat bran fermented with both strains through co-fermentation was labeled as LM-WB.

### 2.4. Determination of Soluble Pentosan Content and Total Phenol Content

The content of soluble pentosan (water-extractable arabinoxylan, WEAX) was determined using the lichenol-hydrochloric acid method [14]. Briefly, 1 mL of supernatant was mixed with 1 mL of 4 mM HCl, 0.3 mL of 1% (*w*/*v*) lichenol, and 3 mL of 1% (*w*/*v*) FeCl_3_. The mixture was incubated in a water bath at 80 °C for 30 min, and the absorbance was measured at 670 nm. The standard curve regression equation for soluble pentosan was y = 1.5069x − 0.0195 (R^2^ = 0.9991).

The total phenol content (TPC) was measured using the Folin–Ciocalteu colorimetric method [15]. In brief, 1 mL of the diluted extract was mixed with 0.5 mL of freshly prepared Folin–Ciocalteu reagent and 3 mL of 20% Na_2_CO_3_, then left to stand for 15 min. The mixture was diluted with 10 mL of distilled water and incubated in the dark at room temperature for 1 h. Absorbance was measured at 760 nm. The TPC was calculated using the regression equation of the gallic acid standard curve: y = 10.946x + 0.0357 (R^2^ = 0.9982).

### 2.5. Determination of Antioxidant Activity

According to Liao’s method [16], 1.0 g of WB was accurately weighed into a 100 mL centrifuge tube, and 25 mL of 70% ethanol was added. The mixture was subjected to ultrasonic extraction at 45 °C for 2 h. The extract was then centrifuged at 5000 rpm for 10 min and stored at 4 °C for antioxidant activity assays.

Antioxidant activity was assessed by measuring DPPH radical scavenging activity, total reducing power, and ABTS^+^ radical scavenging capacity according to established methods [17,18,19].

### 2.6. Solid Phase Microextraction and GC-MS Analysis of Fermented Wheat Bran

Solid-phase microextraction (SPME) coupled with gas chromatography–mass spectrometry (GC-MS) has been widely applied in the analysis of flavor compounds in spices and foods [20]. To evaluate the volatile compounds in fermented WB, GC-MS (GCMS-QP2010 Ultra, Shimadzu, Japan) was used in combination with SPME. The system was equipped with an Rtx-5MS capillary column (30 m × 0.25 mm × 1 μm). Helium served as the carrier gas at a flow rate of 1.0 mL/min, with a split ratio of 1:10. The oven temperature program was as follows: initial temperature of 40 °C (held for 3 min), increased at 4 °C/min to 150 °C (held for 1 min), then at 8 °C/min to 250 °C (held for 6 min). The injector temperature was set at 260 °C. Mass spectra were recorded using electron impact ionization at 70 eV and an ion source temperature of 200 °C. Scanning was performed over an *m/z* range of 35–500.

### 2.7. Statistical Analysis

All assays were performed in triplicate, and results are presented as the mean ± standard deviation. Statistical analyses were conducted using Origin 2021 and SAS 9.2 software. Differences among groups were evaluated using one-way analysis of variance (ANOVA). A *p*-value < 0.05 was considered statistically significant.

## 3. Results

### 3.1. Changes in Soluble Pentosan Content During Fermentation

The soluble pentosan content was expressed as mg/g (WEAX/wheat bran). All fermented samples showed increased WEAX levels compared to unfermented wheat bran, which had a baseline value of 3.60 mg/g (Figure 1). WEAX levels peaked on the fourth day of fermentation. Single-culture fermentation with *L. plantarum* increased WEAX to 42.42 mg/g, an 11.78-fold increase. Co-culture fermentation with *L. plantarum* and *M. purpureus* raised the WEAX content to 45.38 mg/g, a 12.61-fold increase.

### 3.2. Changes in the TPC of Wheat Bran During Fermentation

The Folin–Ciocalteu method was used to determine the TPC, expressed as mg gallic acid equivalents per gram of dry weight (mg GAE/g DW). Compared to unfermented wheat bran, which had a TPC of 1.94 mg/g, all samples except L-WB exhibited increased phenolic contents (Figure 2). On the sixth day, M-WB and LM-WB reached TPC values of 10.29 mg/g and 11.14 mg/g, respectively, representing more than a 5-fold increase.

### 3.3. Selection of the Optimal Fermentation Time

To evaluate the variation in active components—soluble pentosan and TPC (Table 1)—an optimization index was used to determine the optimal fermentation time for subsequent experiments. The optimization index (Y) was defined as: Y = soluble dietary fiber increase rate + total phenol increase rate. The increase rate was calculated as (sample content − control content)/fermentation time. Based on this calculation, co-fermentation of wheat bran with *L. plantarum* and *M. purpureus* in a 1:1 ratio on the second day was identified as the optimal condition for further experiments [20].

### 3.4. Antioxidant Activity of Wheat Bran

To further evaluate the antioxidant activity of fermented wheat bran, ABTS and DPPH radical scavenging assays were performed. Compared to unfermented wheat bran (ABTS and DPPH scavenging rates of 79.82% and 61.73%, respectively), all fermented samples showed enhanced scavenging activity (Figure 3a,b).

The hydroxyl radical scavenging capacity is shown in Figure 3c. The highest scavenging rate was observed in M-WB (95.53%), followed by LM-WB (82.76%), L-WB (74.67%), and C-WB (38.68%).

FRAP reflects the reducing capacity of antioxidants. Under low pH conditions, Fe^3+^-TPTZ (2,4,6-tripyridin-2-yl-1,3,5-triazine) is reduced to the blue-violet Fe^2+^-TPTZ complex. The total antioxidant capacity was measured at 593 nm. A higher absorbance indicates stronger FRAP activity, suggesting a greater ability to scavenge free radicals and prevent oxidative damage. As shown in Figure 3d, all fermented wheat bran samples exhibited improved ion reduction activity, with L-WB, M-WB, and LM-WB showing a concentration-dependent increase.

### 3.5. Determination of Volatile Substances in Fermented Wheat Bran

Aroma is one of the most important sensory attributes of fermented products. Table A1 and Figure 4 show that after fermentation, the concentration of the flavor compounds in the fermented wheat bran changed significantly. A total of 129 volatile compounds were identified across all samples, classified into nine categories: alcohols (fifteen), ketones (twenty-two), acids (seven), esters (eight), aldehydes (sixteen), hydrocarbons (twenty-seven), phenols (three), pyrazines (twenty-two), and others (nine).

The main flavor components of unfermented wheat bran (C-WB) were aldehydes, with a maximum content of 59.46%, followed by ketones at 13.46%. Among the aldehydes, isovaleraldehyde and hexanal were identified at 24.89% and 16.57%, respectively. Isovaleraldehyde exhibited a pungent aroma, while hexanal had a grassy aroma. Octanal showed a sweet orange honey scent. Ketones accounted for 13.46%, with 3-methyl-2-heptanone and 2-heptanone comprising the larger proportion, characterized by a pungent, sharp odor that is often described as solvent- or chemical-like.

The alcohol content of L-WB decreased from 8.52% to 5.13%, ketones decreased from 13.46% to 4.7%, and aldehydes declined from 59.46% to 15.87%. The flavor compounds in wheat bran fermented by *L. plantarum* showed increased hydrocarbons and pyrazines, accounting for 29.04% and 26.32%, respectively. The dodecane content rose from 0.29% in unfermented bran to 6.07%, exhibiting a wine aroma. Among the alcohols, 2,3-butanediol was newly identified at 3.42%, also with a wine aroma. 3-hydroxy-2-butanone was newly identified among ketones at 4.13%, characterized by a strong cream, fat, and pleasant milk aroma. Tetramethylpyrazine, newly identified in pyrazines, contributed fruity and nutty aromas to the fermented bran. Esters also increased significantly; they typically exhibit fruity, fragrant, and sweet smells with low odor thresholds.

The flavor compounds of wheat bran fermented by *M. purpureus* showed higher ketone and alcohol contents, at 31.85% and 26.27%, respectively. The content of 2-decanone increased from 0.18% in unfermented bran to 2%, exhibiting an orange or lemon aroma. Among the alcohols, 3-penten-1-ol was newly identified at 13.09% and is commonly used in flavor production. 2-ethyl-5-methylpyrazine, newly identified among pyrazines at 0.67%, had a nutty or cocoa aroma. Other newly identified compounds included 2-methylfuran, 1-(2-furanylmethyl)-1H-pyrrole, theophylline, and pyrroles, which may impart cream, caramel, and nutty aromas to the fermented bran.

The highest acid and alcohol contents were found in wheat bran fermented using a mixture of *M. purpureus* and *L. plantarum*, accounting for 32.88% and 29.99%, respectively, with other flavor compounds elevated to 9.22%. Among acids, 3-methylbutyric acid and malonic acid were predominant and mainly contributed to the flavor. Among alcohols, 2-methyl-3-buten-1-ol reached 14.96%, characterized by a fermented, oily, and fruity aroma. Pyrazines such as ethyl pyrazine, 2-vinyl-6-methylpyrazine, and 2-ethyl-3,5-dimethylpyrazine, which provide nutty and roasted coffee aromas, are widely used in food flavoring. Additionally, 2-vinylfuran was newly identified among other flavor compounds, imparting bean, fruity, sweet, and nutty aromas.

Principal component analysis (PCA) is a projection method that facilitates visualization of all the information contained in a dataset. Additionally, PCA identifies how samples differ and which variables contribute most to these differences. PCA based on the relative content of volatile classes demonstrated that the aroma profile of wheat bran was significantly influenced by strain combinations (Figure 5). PC1 accounted for 56.0% of the variance, while PC2 explained 33.1%, as clearly shown in Figure 5. Wheat bran fermented by *L. plantarum* exhibited the strongest volatile odor.

## 4. Discussion

### 4.1. Changes in Soluble Pentosan Content During Fermentation

Soluble pentosan, also known as arabinoxylan, is the main dietary fiber component in wheat bran and exhibits biological activities such as improving intestinal flora, lowering blood sugar and lipids, and enhancing immunity [21,22]. Fermentation and acidification of the substrate have been shown to increase soluble pentosan content, consistent with previous studies [10,23]. The fermentation process enhances polysaccharide dissolution by deconstructing the cell wall’s cellulose hemicellulose complex [23]. Meanwhile, the organic acids produced by fermentation metabolism promote a decrease in the pH of the system, which improves the extraction efficiency of WEAX by altering the solubility characteristics of polysaccharides [24].

### 4.2. Changes in the TPC and Antioxidant Activity of Wheat Bran During Fermentation

Wheat bran is rich in phenolic compounds with strong antioxidant activity, which positively affects diabetes, cardiovascular disease, and cancers [25]. Phenolics in wheat bran often exist in bound forms linked by glycosidic and ester bonds, making their extraction difficult. Hydrolytic enzymes produced during fermentation break these covalent bonds, releasing free phenolics and thereby increasing the total phenolic content (TPC). Co-culture fermentation by *L. plantarum* and *M. purpureus* was more effective at releasing free phenolics and increasing TPC in wheat bran. This co-culture system may create favorable conditions for mutualism between the two microorganisms, further facilitating phenolic release [26,27,28]. The phenolic content is the primary source of wheat bran’s antioxidant activity. Generally, the ability of microorganisms to convert bound phenolics into free forms influenced the TPC in fermented samples [29]. Co-culture fermentation by *L. plantarum* and *M. purpureus* showed promising potential for maximizing antioxidant activity, consistent with Tang’s findings [22].

### 4.3. Determination of Volatile Substances in Fermented Wheat Bran

Previous studies reported that most aldehydes decreased during the *L. plantarum* fermentation of cherry, pineapple, carrot, and horse gram sprouts [30,31]. Aldehydes typically exhibit a pronounced fatty odor, but high concentrations may cause off-flavors [32]. However, further research indicated that nonanal, which has a sweet orange and oily aroma, was higher in the aldehyde group of fermented wheat bran (L-WB) compared to unfermented bran (C-WB). The aldehyde content in L-WB was higher than in C-WB. Nonanal may mask unpleasant flavors in the samples [33,34]. Pyrazines are important flavor compounds that significantly influence sensory characteristics due to their unique organoleptic properties [35]. Tetramethylpyrazine, newly identified among pyrazines, imparted fruity and nutty aromas to the fermented bran. Esters also increased significantly; they commonly exhibit fruity, fragrant, and sweet aromas with low odor thresholds and are primary contributors to liquor flavors [36]. These results highlight the potential of *L. plantarum* to enhance ester-like aromas in wheat bran.

Wheat bran fermented by *M. purpureus* showed higher ketone and alcohol contents. Among the alcohols, 3-penten-1-ol, which was newly identified, contributed a flavor and fragrance odor. Pyrazines and methylfuran, newly identified among pyrazines and other compounds, may impart cream, caramel, and nutty aromas. Pyrazine contributes to the development of baking, roasting, and nutty flavors in food [37]. Furans are produced through the degradation of carbohydrates and caramelization reactions, leading to the development of caramel and vegetable flavors [38].

The highest acid and alcohol contents were observed in wheat bran fermented using a mixture of *L. plantarum* and *M. purpureus*, likely due to the synergistic action of *M. purpureus* and enzymes from *L. plantarum* during fermentation. This complex alcohol profile produced a more integrated odor [39]. Among other flavor compounds, 2-vinylfuran was newly identified, imparting bean, fruity, sweet, and nutty aromas [37]. The addition of wheat bran co-fermented by *L. plantarum* and *M. purpureus* offers a viable approach to enhance flavor and impart a distinctive aroma to wheat bran. These findings are consistent with previous reports [21,40].

*L.plantarum* fermentation produced more esters and pyrazines but produced more hydrocarbons. *M. purpureus* fermentation produced more alcohols but produced more ketones. This is not conducive to the release of flavor substances. This study found that co-fermentation significantly altered the flavor compounds of wheat bran compared to single-strain fermentation, characterized by decreased aldehydes and ketones and increased esters, alcohols, and pyrazines, which contributed fruity, sweet, and nutty aromas. It indicated that co-fermentation could significantly impart richer flavor attributes to WB, which is consistent with the previous observation in fermented wheat bran [24]. In addition, many flavor compounds, including L-WB and M-WB, were detected in the LM-WB group during fermentation, indicating that co-fermentation provides the best flavor properties. This is due to the interaction of aldehydes, ketones, esters, and hydrocarbons produced by strains that degrade bran during fermentation. This suggests that solid-state fermentation regulates the bran’s flavor not only through a single substance or its content but also through the synergistic effect of various flavor compounds [40,41,42].

## 5. Conclusions

Fermented wheat bran samples differed markedly from unfermented bran. After fermentation, WEAX, TPC, and antioxidant activity increased, with co-fermentation effectively improving wheat bran quality. Among the strains studied, *M. purpureus* had the greatest impact on these parameters. The flavor profile of wheat bran co-fermented by *L. plantarum* and *M. purpureus* (LM-WB) showed decreased aldehydes and increased alcohols and pyrazines, imparting fruity, sweet, and nutty aromas. Co-fermentation released more functional components and enhanced the flavor properties of wheat bran, providing a scientific basis for its valuable application in the food industry.

The co-fermentation technique significantly improved both the nutritional value and flavor of wheat bran, supporting its high-value utilization in food products. Future studies could further optimize fermentation conditions and explore synergistic effects with other food ingredients to promote broader applications of wheat bran in functional foods. However, co-fermented wheat bran faces challenges in odor control, resulting in significant batch differences regarding volatile substances. At the same time, there are limitations such as insufficient research on the long-term storage stability of aroma substances and the need to optimize the texture fusion degree when blending wheat bran’s rough texture with other food ingredients.

## Figures and Tables

**Figure 1 microorganisms-13-01546-f001:**
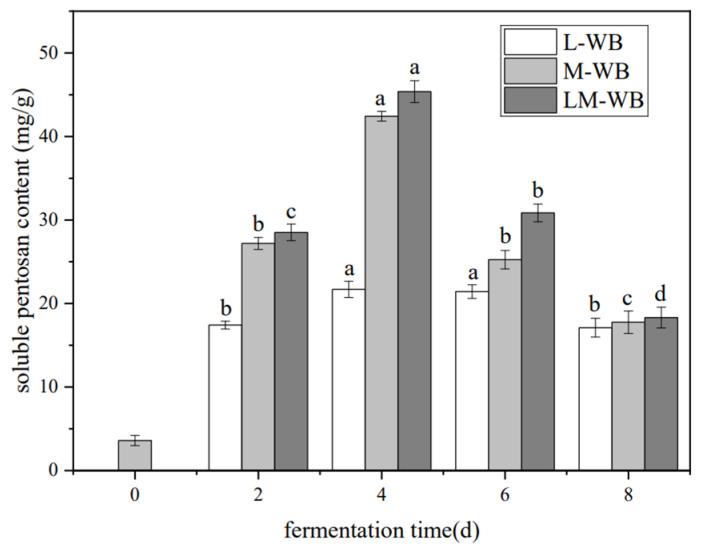
Changes in soluble pentosan content during the fermentation of C-WB, L-WB, M-WB, and LM-WB. Note: Different lowercase letters indicate a significant difference in fermentation time at *p* < 0.05. C-WB is unfermented bran, L-WB is *L. plantarum*-fermented bran, M-WB is *M. purpureus*-fermented bran, and LM-WB is *M. purpureus*/*L. plantarum* = 1:1-fermented bran, as follows.

**Figure 2 microorganisms-13-01546-f002:**
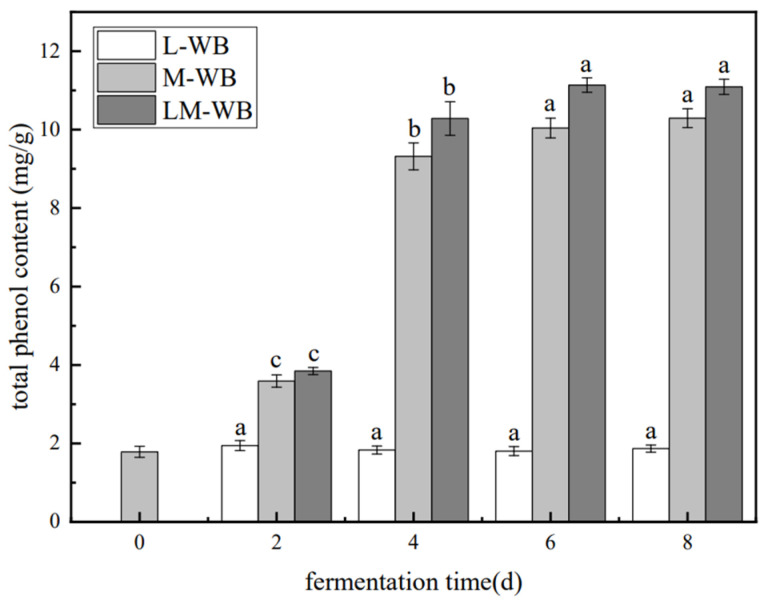
Changes in total phenol content during the fermentation of C-WB, L-WB, M-WB, and LM-WB. Note: Different lowercase letters indicate a significant difference in fermentation time at *p* < 0.05. C-WB is unfermented bran, L-WB is *L. plantarum*-fermented bran, M-WB is *M. purpureus*-fermented bran, and LM-WB is *M. purpureus*/*L. plantarum* = 1:1-fermented bran, as follows.

**Figure 3 microorganisms-13-01546-f003:**
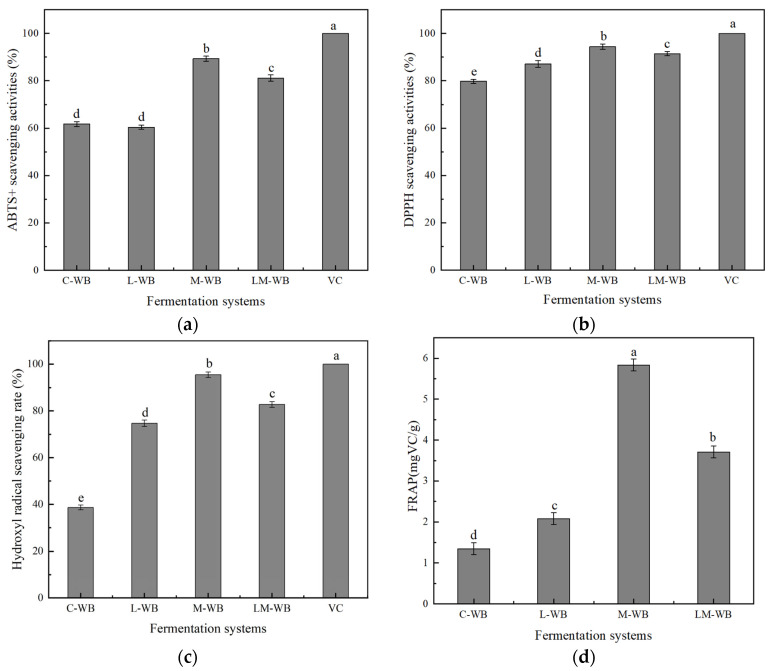
DPPH scavenging activity (a), ABTS scavenging activity (b), hydroxyl radical scavenging rate (c), and FRAP (d) of C-WB, L-WB, M-WB, and LM-WB. Note: Significant differences (*p* < 0.05) were found between the values represented by different letters in the same index.

**Figure 4 microorganisms-13-01546-f004:**
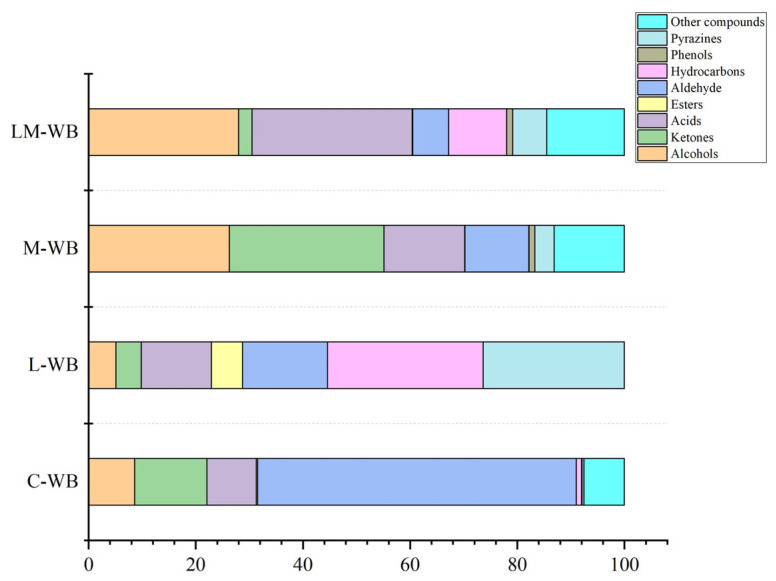
Changes in the relative content of major compound classes in wheat bran fermented by different microbial strains.

**Figure 5 microorganisms-13-01546-f005:**
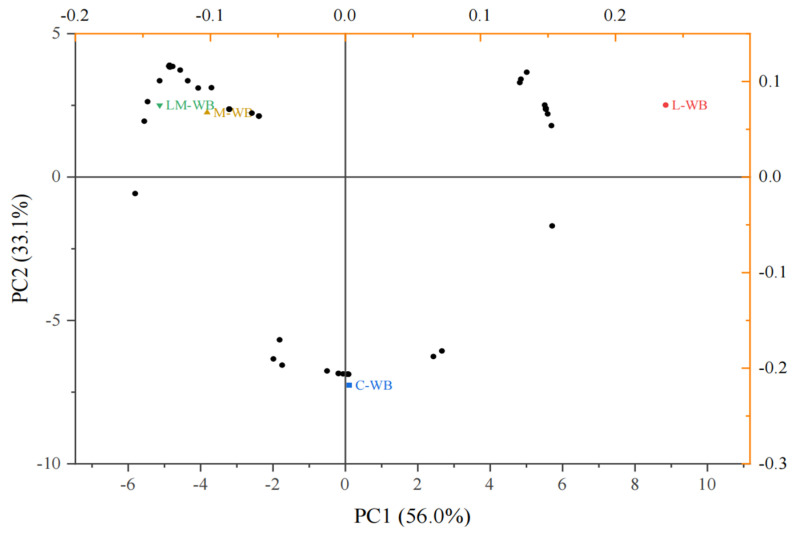
PC1 vs. PC2—scatter plot of the main sources of variability among different wheat bran samples.

**Table 1 microorganisms-13-01546-t001:** Optimization index Y.

Increase Rate	2d	4d	6d	8d
soluble dietary fiber	12.46 ^a^ ± 0.20	10.44 ^b^ ± 0.18	4.54 ^c^ ± 0.08	1.84 ^d^ ± 0.08
total phenol	1.03 ^d^ ± 0.02	2.12 ^a^ ± 0.0.07	1.56 ^b^ ± 0.01	1.16 ^c^ ± 0.01
Y	13.49 ^a^ ± 0.17	12.57 ^b^ ± 0.25	6.10 ^c^ ± 0.08	3.00 ^d^ ± 0.09

Note: Significant differences (*p* < 0.05) were found between the values represented by different letters in the same index.

## Data Availability

The original contributions presented in this study are included in the article. Further inquiries can be directed to the corresponding author.

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
