# Peer review of "Impact of Co-Fermentation on the Soluble Pentosan, Total Phenol, Antioxidant Activity, and Flavor Properties of Wheat Bran"

_microorganisms, 2025, doi:10.3390/microorganisms13071546_

Round 1

Reviewer 1 Report

Comments and Suggestions for Authors
  1. The main objective of the study was to evaluate the effect of co-fermentation on soluble pentosan, total phenol, antioxidant activity and taste properties of wheat bran.
  2. The topic of the research undertaken by the authors is interesting. In order to degrade cellulose in wheat bran to improve taste and aroma, the authors used fermentation by Lactobacillus plantarum and Monascus purpureus - separately and in co-fermentation.
  3. This manuscript contributes new knowledge to this topic compared to other published articles, using co-fermentation of a bacterium (L. plantarum) with a filamentous fungus (M. purpureus).
  4. Notes in the methods section:
  • Line 57: Do Lactobacillus plantarum and Monascus purpureus strains have collection numbers? In manuscripts, species names should be with strain numbers.
  • Lines 62-67, Section 3.2.: This section should discuss the preparation of the four variants of samples, i.e. C-WB, L-WB, M-WB and LM-WB.
  • Lines 62-67, Section 3.2.: The term "seed solution" is unclear and not used in microbiology. It should be: "inoculum".
  • Line 72: The phrase “seed culture broth” is unclear and not used in microbiology. The phrase “inoculum” should be used.
  • Lines 108-109, Section 3.1.: The first sentence about the standard curve should be in section 2.4, not section 3. Results.
  • Lines 124-125, Section 3.2: The sentence about the regression equation of the standard curve should be in section 2.5.
  1. Notes in research results section:
  • Line 136-140: Fig. 1 and Fig. 2 do not show soluble dietary fiber increase rate and total phenol increase rate. Fig. 1 and Fig. 2 show soluble pentosan content and total phenol content. Therefore, the calculation method of the Y index is unclear. Please provide all index values in a separate table and in the methodology section discuss the formula that was used to calculate the optimization index Y.
  • Lines 106-216, Section Results: from the results presented in Fig. 1-3 it does not follow that the coculture of Lactobacillus plantarum and Monascus purpureus (LM-WB) had a better effect on the soluble pentosan content (Fig.1) and total phenol content (Fig.2) and antioxidant activity (Fig.3) than Monascus purpureus fermentation of wheat bran (M-WB). The results only showed that LM-WB influenced the changes in the volatile substance of fermented wheat bran (Fig. 4 – 5 and Table A1).
  1. The discussion is complete and properly discussed with the results of other authors.
  2. Notes in Conclusions Section:
  • Lines 271-276, Section 5. Conclusions: In summary, it should be clearly emphasized that the beneficial effect of co-fermentation concerns only the sensory properties of fermented wheat bran, while the greatest impact on the remaining studied parameters was had by Monascus purpureus (M-WB).
  1. References are adequate. 32 articles were used, 84% of which are from the last 10 years and 63% from last 5 years. This proves the novelty of the topic discussed.
  2. Line 131: The title of Fig.2 lacks a Note, which is only in the title of Fig.1. Please complete.

Reviewer 2 Report

Comments and Suggestions for Authors

The manuscript "Impact of Cofermentation on Soluble Pentosan, Total Phenol, Antioxidant Activity and Flavor Properties of Wheat Bran" presents interesting research findings; however, it requires certain improvements prior to publication.

Specific comments:

  • The abstract should include more detailed results.

  • The current set of keywords is very limited; additional keywords should be included to improve the article’s visibility and indexing.

  • According to the updated nomenclature, the correct name is Lactiplantibacillus plantarum.

  • The Introduction section should be expanded to better introduce the reader to the subject matter of the article.

  • The microbial strains used should be described in more detail (e.g., whether they originate from publicly available culture collections; if they are environmental isolates, whether they have been genetically identified, and whether the DNA sequence fragments are accessible in public databases).

  • The methodology should be described with greater precision to ensure the reproducibility of the experiments.

Reviewer 3 Report

Comments and Suggestions for Authors

The topic addressed in the manuscript is timely and relevant, by addressing the area of efficient utilization of underutilized by-products from the food industry. As the subject of this manuscript is of considerable interest, hence it deserves a more in-depth dissertation.

However, some aspects are recommended to be analyzed and addressed by the authors:

-some keywords associated with the title could be added

-the sentence “Solid phase microextraction and GC-MS was applied to evaluate the flavor” is not necessary to be included in the abstract.

-the presentation of the results within abstract should be similar for all compounds: “the soluble pentosan were 6.03 times 20 higher than in raw bran. Total phenol were 5.74 times higher than in raw bran and antioxidant activity both increased”.

-lines 16-18 – the second and the third sentences should be rather linked / combined.

-line 32 – the bran is not actually enriched; thus, the term should be replaced with a more appropriate one.

-line 37 – … as many bioactive components can be produced by fermentation.

-line 41 – it is recommended to rephrase as “… and it is widely used”.

-lines 59-61 – the sentence is not fully clear.

-paragraphs 2.2 and 2.3 -it is not very clearly explained how the inoculum is prepared and how the individual microorganisms (as later in the manuscript seems the single culture fermentation is performed), as well as the co-culture, are inoculated.

-section 3.3 – the approach is not fully clear; it is recommended to be explained more carefully.

-line 145 - 146 - suggestion for a slight correction: “… 61.73 %, respectively), all fermented wheat bran increased both ABTS and DPPH scavenging rates.”

-line 151 – please detail what FRAP means.

-line 161 – suggestion for slight rephrasing: “Table A1 and Figure 4 show that, after fermentation, the concentration of the flavour compounds in the fermented wheat bran…”.

-section 3.5 - it is not fully clear what the authors mean by “newly detected” associated with the identified compounds.

-lines 168-169 – it is recommended to rephrase the sentence, in order to appropriately illustrate the quantitative aspects of the determinations.

-the Latin names of all microorganisms should be written using Italic characters.

-the last paragraph in section 4.3 should be written more clearly and some corrections should be implemented.

-the advantages of the application of co-culture fermentation should be better highlighted in order to better make the difference between this approach and the fermentation using monocultures (taking into consideration that several better results and advantages of monocultures, such as with L. plantarum, are addressed and pointed out).

-Conclusions should be revised and restructured to improve its clarity and conciseness. In addition, it is recommended to add a clear and concise paragraph summarizing the findings and recommendations for practical applications.

Round 2

Reviewer 2 Report

Comments and Suggestions for Authors

The article has been improved according to the reviewer's comments, I have no additional questions.

Reviewer 3 Report

Comments and Suggestions for Authors

The responses offered by authors and the overall improvement of the manuscript are reasonable and satisfactory.

I consider the paper acceptable for publication in its current form.
